# Intranasal Immunization with Zika Virus Envelope Domain III-Flagellin Fusion Protein Elicits Systemic and Mucosal Immune Responses and Protection against Subcutaneous and Intravaginal Virus Challenges

**DOI:** 10.3390/pharmaceutics14051014

**Published:** 2022-05-08

**Authors:** Chi-Hsun Chen, Chung-Chu Chen, Wei-Bo Wang, Vania Lionel, Chia-Chyi Liu, Li-Min Huang, Suh-Chin Wu

**Affiliations:** 1Institute of Biotechnology, National Tsing Hua University, Hsinchu 30013, Taiwan; cormac6206@gmail.com (C.-H.C.); t810134@gmail.com (W.-B.W.); vania.lionel22@gmail.com (V.L.); 2Department of Internal Medicine, MacKay Memorial Hospital, Hsinchu 30071, Taiwan; 4059@mmh.org.tw; 3Teaching Center of Natural Science, Minghsin University of Science and Technology, Hsinchu 30401, Taiwan; 4National Institute of Infectious Diseases and Vaccinology, National Health Research Institutes, Zhunan 35053, Taiwan; georgeliu@nhri.org.tw; 5Department of Pediatrics, National Taiwan University Hospital, Taipei 10041, Taiwan; lmhuang@ntu.edu.tw; 6Department of Medical Science, National Tsing Hua University, Hsinchu 30013, Taiwan; 7Adimmune Corporation, Taichung 42723, Taiwan

**Keywords:** intranasal immunization, Zika virus, domain III, mucosal vaccine

## Abstract

Zika virus (ZIKV) infections in humans are mainly transmitted by the mosquito vectors, but human-to-human sexual transmission is also another important route. Developing a ZIKV mucosal vaccine that can elicit both systemic and mucosal immune responses is of particular interest. In this study, we constructed a recombinant ZIKV envelope DIII (ZDIII) protein genetically fused with *Salmonella typhimurium* flagellin (FliC-ZDIII) as a novel mucosal antigen for intranasal immunization. The results indicated that the FliC-ZDIII fusion proteins formulated with *E. coli* heat-labile enterotoxin B subunit (LTIIb-B5) adjuvant greatly increased the ZDIII-specific IgG, IgA, and neutralizing titers in sera, and the ZDIII-specific IgA titers in bronchoalveolar lavage and vaginal fluids. Protective immunity was further assessed by subcutaneous and intravaginal ZIKV challenges. The second-generation FliCΔD3-2ZDIII was shown to result in a reduced titer of anti-FliC IgG antibodies in sera and still retained the same levels of serum IgG, IgA, and neutralizing antibodies and mucosal IgA antibodies without compromising the vaccine antigenicity. Therefore, intranasal immunization with FliCΔD3-2ZDIII fusion proteins formulated with LTIIb-B5 adjuvant elicited the greatest protective immunity against subcutaneous and intravaginal ZIKV challenges. Our findings indicated that the combination of FliCΔD3-2ZDIII fusion proteins and LTIIb-B5 adjuvant for intranasal immunization can be used for developing ZIKV mucosal vaccines.

## 1. Introduction

Zika virus (ZIKV) is a small enveloped positive-strand RNA virus belonging to genus *Flavivirus*, family *Flaviviridae* [1]. The recent outbreak of ZIKV infections in Brazil in 2014 [2] was blamed for a 20-fold increase in neonatal microcephaly [3]. The ZIKV epidemic has continued to spread throughout South and Central America [4,5]. The World Health Organization declared it a “public health emergency of international concern” in 2016 [6]. Although the majority of ZIKV infections in humans are transmitted by *Aedes* mosquito vectors, such as *A. aegypti* and *A. albopictus* [7,8], ZIKV transmission through sexual partners is another important route for virus transmission in humans [9,10]. Several recent findings have revealed that ZIKV persisted in patient semen for six months after disease onset [11,12], as well as in the female genital tract [13,14]. The permissive nature of the vaginal mucosa for ZIKV replication, the potential for infection at other sites of the reproductive tract, and the role of progesterone and sex hormones in ZIKV infection susceptibility have also been highlighted for ZIKV transmission [15,16,17]. To date, the most current ZIKV vaccine candidates have been developed using injection or electroporation delivery [18]. Therefore, developing ZIKV mucosal vaccines may provide advantages to elicit both systemic and mucosal immune responses to prevent ZIKV mosquito infections and human-to-human sexual transmission.

In all flavivirus groups, the RNA genome encodes three structural (core C, membrane precursor prM, and envelope E) and seven nonstructural (NS1, NS2A, NS2B, NS3, NS4A, NS4B, and NS5) genes, with flanking untranslated region genes at the 5′- and 3′- ends [1]. Flavivirus E protein is the major protein involved in receptor binding and fusion. The E protein is formed as a head-to-tail dimer on the surface of viral particles. The E monomer consists of three distinct domains in the ectodomain: (i) a central beta-barrel domain I (DI), (ii) an extended finger-like dimerization domain II (DII), and (ii) an immunoglobulin-like domain III (DIII) [19,20]. The DIII of flavivirus E protein is an immunoglobulin-like domain that mainly elicits type-specific antibodies. Some of ZIKV DIII (ZDIII)-specific monoclonal antibodies (mAbs) have been demonstrated to be protective in mice [21,22]. High-resolution X-ray crystal structures have revealed that the Fab and scFb of anti-ZDIII neutralizing mAbs bind to three non-overlapping conformation epitopes: the lateral ridge, C-C’ loop and ABDE sheet regions [21]. Only the lateral ridge-binding mAbs were able to neutralize ZIKV strains and provide passive protection after a live virus challenge [21]. DIII is a promising target for vaccine development. Recombinant ZDIII proteins expressed in *Escherichia coli* (*E. coli*) [23] and plant cells [24] have been reported to induce neutralizing antibodies in mice. Recombinant ZDIII proteins obtained from *Drosophila* S2 insect cells [25] and HEK-293 mammalian cells [26] were reported to elicit neutralizing antibodies and provide protection in newborn or AG129 mice [25,26].

Bacterial flagellin is the major structural component in flagellated bacteria. It has been shown to trigger the TLR5 innate pathways [27,28] and activate intracellular NLRC4 inflammasome-mediated innate responses [29]. Thus, flagellin can be used as a vaccine adjuvant to enhance both systemic and mucosal immunity [30,31]. We have previously shown the use of *E. coli* type IIb heat labile enterotoxin B subunit (LTIIb-B5) as a mucosal adjuvant for intranasal immunizations to enhance protective immunity against H5N1 avian influenza virus infection [32]. In this study, we constructed the recombinant protein of ZDIII fused with *Salmonella typhimurium* flagellin (FliC-ZDIII) and investigated the combination of FliC-ZDIII without or with LTIIb-B5 adjuvant for intranasal vaccination. We characterized both systemic and mucosal antibody responses in sera, bronchoalveolar lavage fluids (BALFs), and vaginal fluids (VFs) for the titers of ZDIII-specific IgG, IgA, and virus neutralizing antibodies. Protective immune responses were determined from immunized BALB/c mice depleted using anti-type I interferon monoclonal antibodies one day before subcutaneous and intravaginal ZIKV challenges, and measured for the survival rate, clinical score, and virus titers in brain tissues and VFs. Our findings may provide useful formation for developing a subunit-based mucosal ZIKV vaccine.

## 2. Materials and Methods

### 2.1. Expression and Purification of Recombinant ZDIII and FliC-ZDIII Fusion Proteins

The cDNA from the ZDIII gene of the Brazil Paraiba_01/2015 strain (GI:1032590576, amino acid residues E589–697) and the fliC gene from *Salmonella typhimurium* (GI:217062, amino acid residues 1–495) were synthesized. These DNAs of ZDIII, FliC-ZDIII, FliCΔD3-2ZDIII, and FliCΔD2ΔD3-3ZDIII were cloned into pET22b (+) plasmid. The constructed plasmids were transformed into *E. coli* BL21 cells (DE3) (Invitrogen) and incubated overnight in 5 mL Luria-Bertani (LB) broth with ampicillin at 37 °C. The *E. coli* culture was transferred to 200 mL of LB broth until the OD_600_ of the cells reached a range of 0.4–0.6, followed by incubation with 1 mM IPTG for 4 h. The cells were centrifuged at 6000× *g* for 10 min at 4 °C. The resulting pellet was resuspended in 40 mL buffer A containing 300 mM Tris, 50 mM NaCl, 10 mM imidazole and 5% glycerol (pH7.2) and was homogenized using an ultrasonic homogenizer at 4 °C. Inclusion bodies were collected via centrifugation at 12,000× *g* for 10 min at 4 °C and then mixed with 8 M urea. The mixture was added to Toyopearl AF-Chelate-650 resin (Sigma Aldrich, St. Louis, MO, USA) and kept overnight, loaded into a column, and washed with 40 mL buffer A containing 0.5% Triton X-100. The recombinant proteins were eluted in 40% buffer B containing 300 mM Tris, 50 mM NaCl, 500 mM imidazole, and 5% glycerol (pH7.2) and dialyzed with phosphate-buffered saline (PBS). The purified proteins were concentrated through a 3 kD centrifugal filter (Sartorius, Göttingen, Germany) and stored at −20 °C. The molecular weight of the purified protein was checked using sodium dodecyl sulfate-polyacrylamide gel electrophoresis (SDS-PAGE) and Coomassie blue staining.

### 2.2. Expression and Purification of Recombinant LTIIb-B5 Protein

The codon-optimized LTIIb-B5 gene (accession No. P43529) was cloned into a pET22b (+) expression vector with a C-terminal 6 His-tag, transformed into *E. coli* BL21 cells (DE3) (Invitrogen, Waltham, MA, USA) grown in Luria-Bertani (LB) broth. Recombinant LTIIb-B5 proteins were produced 4 h following 1 mM IPTG stimulation. Cell pellets were collected and homogenized. Inclusion bodies were solubilized with 8M urea, eluted with 30–40% Buffer B (300 mM Tris, 50 mM NaCl, 500 mM imidazole, 5% glycerol; pH 7.2) by Nickel affinity chromatography column, and dialyzed with 1× PBS at 4 °C overnight. Purified proteins were concentrated using 10 K centrifugal filters (MILLPORE, Burlington, MA, USA), passed through an endotoxin removal column (Cellfine, Tokyo, Japan).

### 2.3. NF-ĸB-Dependent Luciferase Reporter Assay

HEK 293 A cells were co-transfected with 7.5 μg pDUO-hTLR5 plasmid (InvivoGen, San Diego, CA, USA) and 3 μg pGL4.32 [luc2p/NF-ĸB-RE/Hygro] plasmid (Promega, Madison, WI, USA). The transfected cells were seeded at a density of 5 × 10^4^ cells/well in a 96-well plate. The next day, FliC, FliC-ZDIII, FliCΔD3-2ZDIII, and FliCΔD2ΔD3-3ZDIII proteins were serially diluted and incubated with the transfected cells for 5 h at 37 °C. The cells were lysed with Glo-lysis buffer (Promega), and a neolite luciferase substrate (PerkinElmer, Waltham, MA, USA) was added. Luciferase activity was determined using a Victor III microplate reader (PerkinElmer).

### 2.4. Mouse Immunizations

Groups of female BALB/c mice (6–8 weeks old; 5 mice per group) were anesthetized with isoflurane and intranasally immunized with (i) 5 μg ZDIII, (ii) 20 μg ZDIII + 1 μg LT-IIbB5, (iii) 20 μg FliC-ZDIII, (iv) 20 μg FliC-ZDIII + 1 μg LT-IIbB5, (v) 40 μg FliC-ZDIII, (vi) 40 μg FliC-ZDIII + 1 μg LT-IIbB5, (vii) 40 μg FliCΔD3-2ZDIII, (viii) 40 μg FliCΔD3-2ZDIII + 1 μg LT-IIbB5, (ix) 40 μg FliCΔD2ΔD3-3ZDIII + 1 μg LT-IIbB5, (x) 40 μg FliCΔD2ΔD3-3ZDIII + 1 μg LT-IIbB5, and (xi) two doses of 40 μg FliCΔD2ΔD3-3ZDIII + 1 μg LT-IIbB5 primed with one dose of the glycan-masking adenovirus vector Ad-ZprME that can elicit the cross-reactive antibodies for DENV and ZIKV infection enhancements as we reported previously [33], as compared to the PBS-immunized group (negative control). Each group was immunized at weeks 0, 3, and 6. Sera were collected at weeks 2, 5, and 8, while VFs were collected at week 8. The mice were sacrificed at week 8 and the spleen, CLNs and BALFs were collected. Sera were pretreated with heat inactivation at 56 °C for 30 min for complement inactivation. Sera, BALFs and VFs samples were stored at −20 °C. The spleen and CLNs were stored at −80 °C. BALB/C mice were provided by National Laboratory Animal Center and kept in plastic see-through cages. Each cage contains a water bottle, feeder and corn cob bedding. Environment condition was based on IACUC policy (Condition: 07~19 light and 19~07 dark; Temp 24 °C). Mice were euthanized after the completion of experiments. Pain of animals were minimized by agents (Ketoprofen). Sacrifice of mice was carried out by CO_2_ inhalation. All experiments were conducted following the compliance of the international legislation concerning the 3 Rs in the animal experimentation in accordance with the guidelines of the Laboratory Animal Center of the National Tsing Hua University (NTHU). Animal use protocols were reviewed and approved by the NTHU Institutional Animal Care and Use Committee (approval no. 10533).

### 2.5. ELISA IgG and IgA Titer Assay

Enzyme-linked immunosorbent assay (ELISA) was used to measure ZDIII-specific IgG and IgA in mouse sera, BALFs and VFs. ELISA plates were coated overnight with 2 μg/mL ZDIII proteins at 4 °C and blocked with 1% bovine serum albumin (BSA) at room temperature (RT) for 1 h. Serial dilutions of the serum samples were added to the ELISA plates and incubated for 1 h at RT. After washing three times with PBS containing 0.05% Tween 20 (PBST), horseradish peroxidase (HRP)-conjugated anti-mouse IgG (1:30,000, Genetex, Irvine, CA, USA), or IgA antibodies (1:10,000, Bethyl, Montgomery, TX, USA) were added. The plates were washed with PBST, developed with TMB substrate (BioLegend, San Diego, CA, USA), and the reaction was stopped with 2N H_2_SO_4_. The absorbance was measured using an ELISA reader at 450 nm. The end-point titers were measured as the four-fold absorbance of the negative control. BALF total IgA was measured using an IgA Mouse Uncoated ELISA Kit (Invitrogen, Waltham, MA, USA) in accordance with the manufacturer’s instructions.

### 2.6. Plaque Reduction Neutralization Test (PRNT)

Vero cells were seeded overnight at 5 × 10^5^ cells/well in 6-well plates at 37 °C. Two-fold serial dilutions of mouse serum samples were incubated with 100 plaque-forming units (PFUs) of ZIKV (PRVABC59, ATCC ^®^VR-1843) for 1 h at 37 °C and then added to 6-well plates. After incubation for 1 h at 37 °C, the infected cells were overlaid with 12% methylcellulose in MEM and incubated at 37 °C for 5–7 days. The cells were fixed and stained with staining buffer containing 2% formalin and 1% crystal violet. The plaques were counted, and the neutralizing antibody titers were determined as the serum dilution causing 50% reduction in plaque number (PRNT_50_).

### 2.7. T Cell Cytokine ELISA Assay

Splenocytes were seeded at 5 × 10^6^ cells/well and 10^6^ cells/well, respectively, in 24-wells plates, and then treated with 10 μg/mL recombinant ZDIII proteins for stimulation for 72 h. The culture supernatants were collected and measured for IFN-γ, IL-4, and IL-17A, and IL22 titers using an ELISA MAX kit (BioLegend) according to the manufacturer’s instruction (BioLegend). The diluted supernatants were incubated with the capture antibodies for 2 h at RT. The cytokines were detected using specific antibodies for 1 h and interacted with avidin-HRP for 30 min prior to determining coloration and end-point titers.

### 2.8. Subcutaneous Virus Challenge

Immunized mice received 2 mg of MAR1-5A3 (IFNAR1-blocking mAb) via the intraperitoneal route one day before virus challenge with 10^8^ PFU of ZIKV via the subcutaneous route. The survival rate and clinical scores were recorded for 14 days. Clinical scores were calculated as follows: normal = 0; ruffled fur = 2; lethargy, pinched, hunched, wasp-waisted = 3; labored breathing, rapid breathing, inactive, neurological = 5; and dead = 10 [34,35]. For the measurement of virus titer in the brain, the challenged mice were sacrificed on day 6, and the brain samples were collected.

### 2.9. Intravaginal Virus Challenge

Immunized mice were administered with 2 mg of medroxyprogesterone acetate (DMPA) 5 days before the virus challenge, and 2 mg of MAR1-5A3 (IFNAR1-blocking mAb) via the intraperitoneal route one day before the virus challenge with 10^8^ PFU of ZIKV via the subcutaneous route. The survival rate of the virus challenged mice was recorded for 14 days. On days 1, 3, 5, and 7, the VFs of the mice were collected by pipetting 20 μL of PBS into the vagina of the mice. ZIKV virus titers of the samples were determined via plaque assay.

### 2.10. Statistical Analysis

Statistical analyses were performed using the GraphPad Prism (GraphPad Software, Inc., San Diego, CA, USA). The statistical significance of differences between the groups was assessed using one-way analysis of variance (ANOVA) with Tukey’s or Holm–Sidak multiple comparison tests. Differences with a *p*-value of less than 0.05 (*), 0.01 (**), and 0.001 (***) were considered statistically significant.

## 3. Results

### 3.1. Expression, Purification, and Characterization of ZDIII and FliC-ZDIII Fusion Proteins

The cDNAs from the ZDIII gene of the Brazil Paraiba_01/2015 strain (GI:1032590576) and the flagellin gene of *Salmonella typhimurium* (GI:217062) were synthesized and constructed for encoding recombinant protein expression in *E. coli* (Figure 1A). ZDIII and FliC-DIII recombinant proteins were purified using a nickel-chelating affinity column with Toyopearl AF-Chelate-650 resin. The purified recombinant proteins were analyzed using SDS-PAGE gels stained with Coomassie blue (13 kDa for ZDIII, 64 kDa for FliC-ZDIII) (Figure 1B). The TLR5 activities of ZDIII and FliC-ZDIII recombinant proteins were analyzed in HEK-293A cells that had been transfected with a human TLR5-expression vector and a luciferase reporter vector. The results showed that recombinant FliC and FliC-ZDIII proteins triggered a dose-dependent increase in luciferase reporter TLR-5 activity in 293A cells (Figure 1C).

### 3.2. Intranasal Immunization with ZDIII and FliC-ZDIII Fusion Proteins plus LTIIb-B5 Mucosal Adjuvant

To investigate the immune responses elicited by intranasal administration of ZDIII and FliC-ZDIII, we incorporated the LTIIb-B5 mucosal adjuvant [32,36,37] into the immunization groups. Groups of BALB/c mice (n = 5 per group) were intranasally immunized with (i) 5 μg ZDIII, (ii) 5 μg ZDIII + 1 μg LT-IIbB5, and (iii) 20 μg FliC-ZDIII, and (iv) 20 μg FliC-ZDIII + 1 μg LT-IIbB5, as compared to the PBS-immunized group (negative control) and the Ad-ZprME immunized group (Figure 2A). The antigen content of 20 μg FliC-ZDIII contained 4 μg ZDIII according to mass conversion (ZDIII = 13 kDa m. wt.; FliC-ZDIII = 64 kDa m. wt.). Three-dose intranasal immunizations were conducted in a three-week interval with or without 1 μg LTIIb-B5 adjuvant, and antisera and mucosal fluids (BALFs and VFs) were collected for measuring antibody responses and T cell responses in the spleen.

Serum samples were collected at two weeks after the first, second, or third dose immunization and analyzed for the titers of ZDIII-specific IgG and IgA antibodies using ELISA. The results showed that the titers of IgG antibodies and IgA antibodies increased by booster immunizations in the second- and third-dose immune sera (Figure 2B,C). In the first-dose sera, IgG and IgA titers were barely detected for immunizations with FliC-ZDIII and FliC-ZDIII + LTIIb-B (Figure 2B,C). Immunizations with the FliC-ZDIII fusion proteins with or without the use of LTIIb-B5 adjuvant greatly increased the elicitations of ZDIII-specific IgG and IgA titers in the second-or third-dose sera. However, no significant differences were observed between FliC-ZIII and FliC-ZDIII + LTIIb-B5 groups. We also measured the neutralizing antibodies (PRNT_50_ titers, log2 scale) in the third dose sera. The results indicated that intranasal immunization with ZDIII + LTIIb-B, FliC-ZDIII, and FliC-ZDIII + LTIIb-B5 elicited detectable neutralizing antibodies (PRNT_50_ titer), but these titers did not show any statistical differences among all groups (Figure 2B).

Mucosal vaccination not only triggers the systemic immune responses, such as presence of neutralizing antibodies in sera, but also induces the mucosal immune responses (i.e., presence of secretory IgA) in mucosal fluids. BALFs and VFs were collected after third dose immunizations and analyzed for ZDIII-specific IgA titers using ELISA. The results indicated that intranasal immunization with FliC-ZDIII + LTIIb-B5 elicited approximately 1.5-log IgA titers in BALFs (Figure 2D) and 1.5-to 2.0-log IgA titers in VFs (Figure 2E). To further confirm the production secretory IgA in mucosal fluids, we measured the overall total IgA titers. Our results indicated that the differences in the amount of overall IgA titer was significant for the FliC-ZDIII and FliC-ZDIII + LTIIb-B5 immunized groups (Figure 2D). Immunizations with FliC-ZDIII and FliC-ZDIII + LTIIb-B5 had higher titers of ZDIII-specific IgG antibodies in BALFs than other groups (Figure 2D). The FliC-ZDIII + LTIIb-B5 immunized group was able to elicit ZDIII-specific IgA antibodies in VFs in mice (Figure 2E).

To measure T cell responses elicited by intranasal immunizations, the spleens were collected three weeks after the third-dose immunization, seeded in 24-well plates, and then incubated with recombinant ZDIII proteins for stimulation for 72 h. The culture supernatants were collected and measured for IFN-γ, IL-4, IL17A, and IL-22 production using ELISA. The results indicated that the level of IFN-γ cytokine production in the spleen of the FliC-ZDIII + LTIIb-B5 group was significantly higher than those in the ZDIII, ZDIII + LTIIb-B5, and FliC-ZDIII, groups (Figure 3A). However, the FliC-ZDIII and FliC-ZDIII + LTIIb-B5 immunized groups showed an increased but still relatively low level of IL-4 production, and no detectable levels were found for the ZDIII and ZDIII + LTIIb-B5 groups (Figure 3B). For IL-17A and IL-22 production, we found that FliC-ZDIII and FliC-ZDIII + LTIIb-B5 had significant titers in the spleen compared to all other immunized groups (Figure 3C,D). Therefore, intranasal immunization with FliC-ZDIII fusion protein induced potent Th1 and Th17 responses.

### 3.3. Protective Immunity in Immunized Mice by Subcutaneous and Intravaginal ZIKV Challenges

To further assess the protective immunity, the same intranasal immunization regimens were conducted in groups of BALB/c mice intranasally immunized with three doses of the FliC-ZDIII antigen at 40 μg per dose for 6 weeks, and the immunized mice were challenged at week 9 with 10^8^ PFU of ZIKV (PRVABC59 strain) via the subcutaneous or intravaginal route. All of the immunized BALB/c mice received 2 mg MAR1-5A3 (IFNAR1-blocking mAb) by intraperitoneal injection one day before the virus challenge at week 9 (Figure 4A). For intravaginal challenges, an additional injection of 2 mg of medroxyprogesterone acetate (DMPA) was administered via the subcutaneous route four days before the virus challenge (Figure 4A). The antisera, BALFs, and VFs after the third dose but before the virus challenge indicated that both the FliC-ZDIII and FliC-ZDIII + LTIIb-B5 groups elicited increased titers of anti-ZDIII IgG, IgA and neutralizing antibodies in sera and IgA antibodies in BALFs and VFs as compared to the PBS-immunized group (Figure 4B,C). For the subcutaneous virus challenge, the survival rate recorded at 14 days post-challenge was 60% (FliC-ZDIII), 33% (FliC-ZDIII + LTIIb-B5), and 20% (PBS) (Figure 4D). The number of the immunized mice in the FliC-ZDIII + LTIIb-B5 group was n = 3 due to the loss of two mice during the second and the third dose immunization periods before the subcutaneous virus challenge, thus resulting in a 33% survival rate (Figure 4D). However, the survival rate has no statistically significant difference among these three groups (Figure 4D). The clinical scores recorded for 14 days after the subcutaneous virus challenge were relatively lower for both the FliC-ZDIII and FliC-ZDIII + LTIIb-B5 groups than in the PBS group (Figure 4D). For the intravaginal challenge, the survival rate was recorded at 14 days after the virus challenge, and VFs were collected on one, three, and five days after virus inoculation and virus titers were measured. The survival rates for all groups were 100% at 14 days after the virus challenge (data not shown). However, the virus titers in VFs were significantly lower for the FliC-ZDIII + LTIIb-B5 group, followed by the FliC-ZDIII group, than the PBS group, one, three, and five days after virus inoculation (Figure 4E).

### 3.4. Immune Responses Elicited by the Second-Generation FliCΔD3-2ZDIII and FliCΔD2ΔD3-3ZDIII Fusion Proteins to Reduce FliC-Specific Adaptive Response

We further constructed the second-generation FliC-ZDIII fusion proteins by replacing FliC D3 with another ZDIII (FliCΔD3-2ZDIII) or replacing FliC D2 and D3 with two repeats (2x) of ZDIII (FliCΔD2ΔD3-3ZDIII) (Figure 5A), as compared to FliC-ZDIII as the first generation antigen. All these fusion proteins were expressed in *E. coli* BL21 and purified using nickel-chelated affinity chromatography. The purified recombinant proteins were eluted and examined using SDS-PAGE gels, all showing single major bands for FliC-ZDIII (m. wt. = 64 kDa), FliCΔD3-2ZDIII (m. wt. = 67 kDa), and FliCΔD2ΔD3-3ZDIII (m. wt. = 66 kDa) in SDS-PAGE gels stained with Commassie blue (Figure 5B). Both FliCΔD3-2ZDIII and FliCΔD2ΔD3-3ZDIII proteins were found to have lowered the dose–response curves for TLR5 activity, and the peak values for the FliCΔD2ΔD3-3ZDIII protein were significantly lower than other groups (Figure 5C).

The same immunization regimen was conducted for these two second-generation antigens, 40 μg FliCΔD3-2ZDIII and FliCΔD2ΔD3-3ZDIII without or with 3 μg LTIIb-B5 adjuvant. The results indicated that the anti-ZDIII IgG titers in the sera for the FliCΔD3-2ZDIII and FliCΔD2ΔD3-3ZDIII groups without or with LTIIb-B5 adjuvant were reduced (not statistically significant) compared the FliC-ZDIII immunized groups (Figure 6A). The anti-ZDIII IgA titers in sera had almost the same titers for all immunized groups except for FliCΔD2ΔD3-3ZDIII + LTIIb-B5 (Figure 6A). For serum neutralizing antibodies, we found that the PRNT50 titer of the FliCΔD2ΔD3-3ZDIII + LTIIb-B5 group was reduced compared to the FliC-ZDIII and FliCΔD3-2ZDIII groups with LTIIb-B5 adjuvant (Figure 6A). Even without the use of LTIIb-B5 adjuvant, the PRNT50 titer of the FliCΔD2ΔD3-3ZDIII group was lower than that of the FliC-ZDIII + LTIIb-B5 and FliCΔD3-2ZDIII + LTIIb-B5 groups (Figure 6A). We also determined the anti-FliC antibodies in the sera after three-dose intranasal immunizations, and all the results followed the order of FliC-ZDIII > FliCΔD3-2ZDIII > FliCΔD2ΔD3-3ZDIII with or without the use of LTIIb-B5 adjuvant (Figure 6B). For mucosal IgA production, we measured the anti-ZDIII IgA titers in BALFs and VFs. Results from all FliC-ZDIII, FliCΔD3-2ZDIII and FliCΔD2ΔD3-3ZDIII immunized groups without or with LTIIb-B5 adjuvant showed no significant differences in IgA titers in VFs (Figure 6C). In BALFs, the anti-ZDIII IgA titers of the three antigens with the use of LTIIb-B5 adjuvant were slightly higher than the titers without LTIIb-B5 adjuvant but no significant differences were observed among FliC-ZDIII, FliCΔD3-2ZDIII and FliCΔD2ΔD3-3ZDIII antigens (Figure 6C). The immunized mice were further intravaginal challenged with ZIKV, and the results showed that the virus titers in VFs on days 1, 3 and 5 were significantly lower for the FliC-ZDIII, FliCΔD3-2ZDIII, and FliCΔD2ΔD3-3ZDIII groups than the PBS control group (Figure 6D). Therefore, the second-generation FliCΔD3-2ZDIII and FliCΔD2ΔD3-3ZDIII antigens resulted in a reduced titer of anti-FliC IgG antibodies in sera but still retained the same titers of serum IgG, IgA, and neutralizing and mucosal IgA antibodies and protective immunity

### 3.5. Intranasal Immunization with the Use of LTIIb-B5 Adjuvant for the Second-Generation FliCΔD3-2ZDIII Fusion Proteins

We further used LTIIb-B5 adjuvant formulated with the second-generation FliCΔD3-2ZDIII fusion protein to enhance mucosal immunity. Three immunized groups (PBS, FliCΔD3-2ZDIII, and FliCΔD3-2ZDIII + LTIIb-B5) were administered three doses of intranasal immunization at weeks 0, 3, and 6, and all of the immunized mice were challenged with ZIKV virus by the subcutaneous or intravaginal route at week 9. Intranasal FliCΔD3-2ZDIII immunization with the use of LTIIb-B5 adjuvant increased the anti-ZDIII IgG, IgA, and neutralizing antibodies in sera (Figure 7A). Moreover, the FliCΔD3-2ZDIII + LTIIb-B5 group elicited a significantly higher titer of anti-ZDIII IgA in VFs (Figure 7B). Therefore, the use of LTIIb-B5 adjuvant with the FliCΔD3-2ZDIII fusion protein for intranasal immunization improved systemic and mucosal immune responses. The protection was further assessed in these immunized mice following the subcutaneous and intravaginal challenges. For the subcutaneous virus challenge, the survival rates for the FliCΔD3-2ZDIII + LTIIb-B5 group was 100% compared to 60% for the FliCΔD3-2ZDIII group and 40% for the PBS group (Figure 7C). The virus titers in brain tissue of the FliCΔD3-2ZDIII + LTIIb-B5 group at six days post-infection were significantly lower than those in the FliCΔD3-2ZDIII and PBS groups (Figure 7D). Notably, the numbers of immunized mice (n = 5 per group) six days after subcutaneous virus challenge dropped to n = 2 for the PBS group and dropped to n = 3 for the FliCΔD3-3ZDIII group. For the intravaginal virus challenge (the survival rates for all immunized groups were 100%), the virus titers in VFs at five days post challenge were lower for the FliCΔD3-2ZDIII and FliCΔD3-2ZDIII + LTIIb-B5 groups compared to the PBS group (Figure 7E). Therefore, intranasal immunization with three doses of FliCΔD3-2ZDIII + LTIIb-B5 was found to elicit systemic and mucosal anti-ZIKV immunity and protection against subcutaneous and intravaginal virus challenges.

## 4. Discussion

Intranasal administration is an effective mucosal vaccine delivery route to elicit antigen-specific systemic and mucosal immune responses [38,39,40]. Antigens can be recognized in nasopharynx-associated lymphoid tissues that contain M cells, antigen-presenting cells, T cells, and B cells to trigger mucosal immune responses such as through IgA-secreting B cells (or plasma cells) [38,39,40]. In this study, we constructed FliC-ZDIII fusion proteins for intranasal immunization and the results demonstrated the elicitation of both systemic and mucosal responses and protection against subcutaneous and intravaginal ZIKV challenges.

We previously reported the success for expression of the FliC and four-serotype dengue virus (DENV1-4) DIII fusion proteins in *E. coli* and showed that the FliC-DENV2 DIII fusion protein enhanced the neutralizing antibody titers up to 100 folds than the titer obtained from the mixture of FliC and DENV2 DIII alone [32]. Direct fusion of FliC with ZDIII provides the advantage for a simultaneous delivery of antigen and adjuvant together to the same antigen-presenting cells, eliminates the FliC protease sensitivity and tendency to aggregate, and induces a strong affinity for TLR5 receptors to trigger more localized immune stimulations [41]. The FliC molecule consists of four globular domains (D0, D1, D2, and D3): the D0 domain contains the N’-terminal and C-terminal elements to trigger the NLRC4 inflammasome [42,43]; the D1 domain contains a critical amino acid residue from three helices to interact with TLR5 [44,45,46]; the hypervariable D2 and D3 domains are immunodominant and can be deleted to abrogate intrinsic antigenicity of flagellin without affecting adjuvant stimulatory activity [47]. In this study, we constructed the second-generation antigens of FliCΔD3-2ZDIII (by deleting FliC D3 and replacing with 1 × ZDIII) and FliCΔD2ΔD3-3ZDIII (by deleting FliC D2 and FliC D3 and replacing with 2 × ZDIII) (Figure 5). Our results indicated that immunization with only the second-generation FliCΔD3-2ZDIII antigen resulted in a reduced titer of anti-FliC IgG antibodies in the sera (Figure 6B) and still retained the same levels of serum IgG, IgA, and neutralizing antibodies and mucosal IgA antibodies (Figure 6A,C). As the bacterial flagellin is an antigen commonly expressed by commensal and pathogenic bacteria in the guts, the flagellin-specific adaptive immunity may link to the development of chronic relapsing intestinal inflammation of the inflammatory bowel diseases such as Crohn’s disease and ulcerative colitis [48,49,50]. Therefore, the second-generation FliCΔD3-2ZDIII antigen can reduce FliC-specific adaptive response without compromising the vaccine antigenicity.

Groups of BALB/c mice were immunized intranasally with FliC-ZDIII fusion proteins for a three-dose regimen in a three-week interval. The titers of anti-ZDIII IgG and IgA antibodies were detected in the first-dose sera, and gradually increased to higher values in the second and third dose sera from the groups immunized with FliC-ZDIII and FliC-ZDIII + LTIIb-B5 as compared to the relatively low titers detected in the groups immunized with ZDIII and ZDIII + LTIIb-B5 (Figure 2B,C). The results also demonstrated titers of the ZDIII-specific IgG and IgA antibodies in BALFs and VFs only after the third-dose immunization for the groups immunized with FliC-ZDIII and FliC-ZDIII + LTIIb-B5 (Figure 2D,E). Since mucosal vaccinations generally require multiple booster doses to induce effective immune responses, three-dose intranasal regimens were reported using influenza hemagglutinin proteins incorporated into chitosan nanoparticles [51,52] or formulated with recombinant flagellin in oil-in-water emulsions [53]. Our studies showed that three-dose intranasal immunization was required to elicit potent anti-ZIKV antibody responses in the sera, BALFs, and VFs. We conducted the three-dose regimen for intranasal immunizations with 20 μg FliC-ZDIII without or with 1 μg LT-IIbB5 (Figure 2). We further increased the antigen dose to 40 μg FliC-ZDIII and the amount of 3 μg LTIIb-B5 adjuvant to assess the protection (Figure 4). The resulted showed an approximately six-fold difference in neutralizing antibody titers (PRNT-50 values) in the third dose sera in between groups of mice immunized with FliC-ZDIII or FliC-ZDIII + LTIIb-B5. However, the IgG and IgA titers in the third dose sera were similar in between groups of mice immunized with FliC-ZDIII or FliC-ZDIII + LTIIb-B5. Compared to the results that we previously reported using FliC-ZDIII for two-dose intramuscular immunization or the first dose adenovirus vector-priming and the second dose FliC-ZDIII booster in BALB/c mice (33), the three-dose intranasal immunization with FliC-ZDIII and LTIIb-B5 adjuvant elicited higher titers of neutralizing antibodies in sera (Figure 2B, Figure 4B, Figure 6A, and Figure 7A). Therefore, intranasal immunization with FliC-ZDIII fusion protein for a three-dose regimen can elicit a comparable systematic immune response similarly to intramuscular immunization. To our knowledge, this is the first report using mucosal delivery of ZIKV recombinant vaccines. Intranasal immunizations with the FliC-ZDIII fusion proteins induced potent Th1 and Th17 responses as compared to the ZDIII proteins alone (Figure 3). The results indicated that the FliC-ZDIII fusion proteins even without the use of LTIIb-B5 adjuvant elicited approximately the same Th17 cellular responses for IL-17A and IL-22 production in the spleen (Figure 3C,D). Mucosal delivery of antigens with flagellin adjuvant has been reported to stimulate TLR5 signaling in dendritic cells to activate processing, presentation, co-stimulatory function and secretion of IL-12, IL-23, or IL-6, thereby promoting the activation of Th1 and Th17 cellular and antibody-mediated immunity as well as innate lymphoid cells [54,55,56]. Flagellin was found to activate TLR-5 signaling in airway epithelial cells, and to trigger proinflammatory responses to recruit neutrophils and dendritic cells to mucosal sites [57] or recapitulate the transcriptional signature of lung responses [58]. The adjuvant potential of flagellin by intranasal delivery was dependent on TLR-5 but not on NLRC4 in radioresistant lung cells [57]. However, a recent finding using MVA vaccinia vector encoding flagellin demonstrated that flagellin can induce NLRC4 inflammasome response to enhance secretory IgA production in the lung and intestinal mucosa [59]. Additional investigations are required to understand the modes of action for the FliC-ZDIII fusion proteins to trigger mucosal immune response. Moreover, we did not measure the CTL response elicited by intranasal immunizations with FliC-ZDIII plus LTIIb-B5 adjuvant in the present study. Further examination is needed to determine the cell-mediated immunity may also contribute to an effective protective immune response.

We assessed the protective immunity using the immunocompetent BALB/c mice administered IFNAR-1-blocking mAb before subcutaneous or intravaginal challenge with ZIKV (Figure 4A). This IFNAR1-blocking model can retain a competent immune system in the immunized mice to investigate the elicited immunity for protection, rather than the use of the immune-incompetent mice such as Ifnar1−/− mice or AG129 mice (interferon-α/β and -γ receptor-knock-out mice) [60,61,62]. Our results indicated that intranasal immunization with FliC-ZDIII and FliCΔD3-2ZDIII fusion proteins protected from subcutaneous live virus challenges, resulting in a 60% survival rate and relatively lower clinical scores (Figure 4D and Figure 7C). We also investigated the protection of the FliC-ZDIII and FliCΔD3-2ZDIII immunized mice through intravaginal challenge, by treating mice with DMPA to synchronize the mouse estrus cycle at a prolonged diestrus phase, rendering mice susceptible to vaginal virus infection [63]. Our results showed that immunization with FliC-ZDIII and FliCΔD3-2ZDIII fusion proteins lowered the viral load of ZIKV titer in vaginal washes on day 5, as compared to the control group (Figure 4E and Figure 7D).

## 5. Conclusions

In this study, we constructed FliC-ZDIII fusion protein as a mucosal vaccine candidate and characterized for its elicitation of ZDIII-specific IgG, IgA, and virus neutralizing antibodies in sera, BALFs, and VFs. We further constructed the second-generation FliCΔD3-2ZDIII fusion protein to reduce the elicitation of anti-FliC IgG antibodies in sera without compromising the vaccine antigenicity. Intranasal immunization with the second-generation FliCΔD3-2ZDIII fusion proteins formulated with LTIIb-B5 elicited the greatest protective immunity against subcutaneous and intravaginal ZIKV challenges. Our findings indicated that the combination of FliCΔD3-2ZDIII fusion proteins and LTIIb-B5 adjuvant for intranasal immunization can be a viable candidate for developing ZIKV mucosal vaccines.

## Figures and Tables

**Figure 1 pharmaceutics-14-01014-f001:**
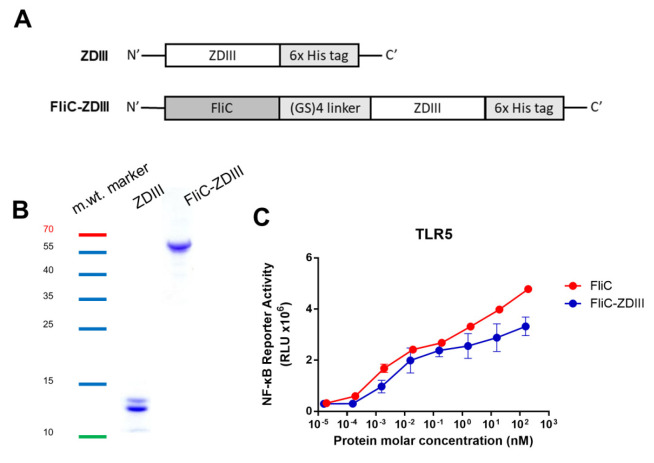
**Expression and characterization of ZDIII and FliC-ZDIII recombinant proteins.** (**A**) ZDIII and FliC-ZDIII, which were linked by a (GS)_4_ linker sequence (GSGSGSGS), were constructed into the pET-22b (+) vector with a C-terminal His-tag for recombinant protein expression. (**B**) ZDIII and FliC-ZDIII recombinant proteins produced by *E. coli* and purified using nickel-chelated affinity chromatography. Purified proteins were verified by SDS-PAGE staining with Coomassie blue. The molecular weight marker is indicated as the color ladder. (**C**) TLR5 signal functional assay was performed by co-culturing the 10-fold serial dilutions of FliC and FliC-ZDIII recombinant proteins with HEK 293A expressing TLR5 receptor and NF-κB reporter vector. The cells were disrupted and treated with neolite luciferase substrate. The TLR5 activity was measured by luciferase activity.

**Figure 2 pharmaceutics-14-01014-f002:**
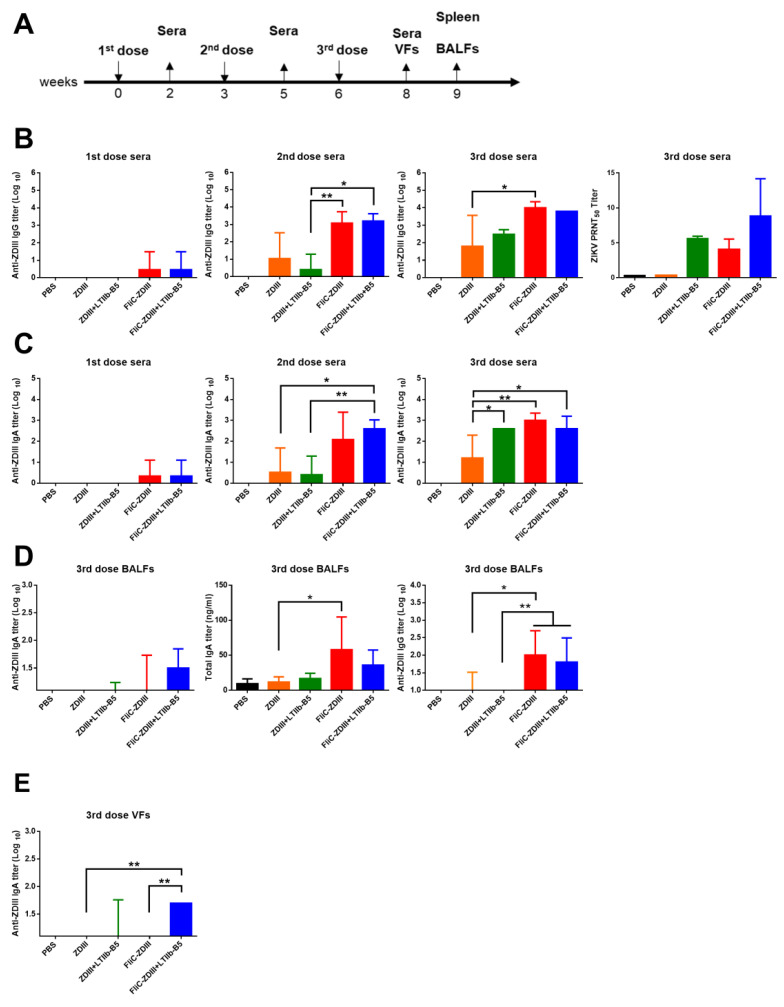
**Immunization with ZDIII and FliC-ZDIII proteins and antibody responses in sera, BALFs, and VFs.** (**A**) Illustration of immunization regimens. Five groups of BALB/c mice (n = 5 per group) of PBS, 5 μg ZDIII, 20 μg ZDIII + 1 μg LTIIb-B5, 20 μg FliC-ZDIII, and 20 μg FliC-ZDIII + 1 μg LTIIb-B5 were immunized three doses via the intranasal route. Sera were collected two weeks after each dose immunization. VFs were collected two weeks after the 3rd dose immunization. The spleen and BALFs were collected three weeks after the 3rd dose immunization; (**B**) ZDIII-specific IgG titers in 1st dose, 2nd dose, and 3rd dose sera; and the ZIKV neutralizing antibody titers (PRNT-50) in 3rd dose sera; (**C**) ZDIII-specific IgA titers in 1st dose, 2nd dose, and 3rd dose sera; (**D**) ZDIII-specific total IgA titers, the overall total IgA titers, and ZDIII-specific total IgG titers in BALFs after the 3rd dose immunization; (**E**) ZDIII-specific total IgA titers in VFs after the 3rd dose. Statistical test for multiple comparison for all groups except PBS were analyzed using one-way ANOVA with Tukey’s or Holm-Sidak’s multiple comparison tests. (*, *p* < 0.05 and **, *p* < 0.01). Error bars are plotted as standard deviation from the mean value.

**Figure 3 pharmaceutics-14-01014-f003:**
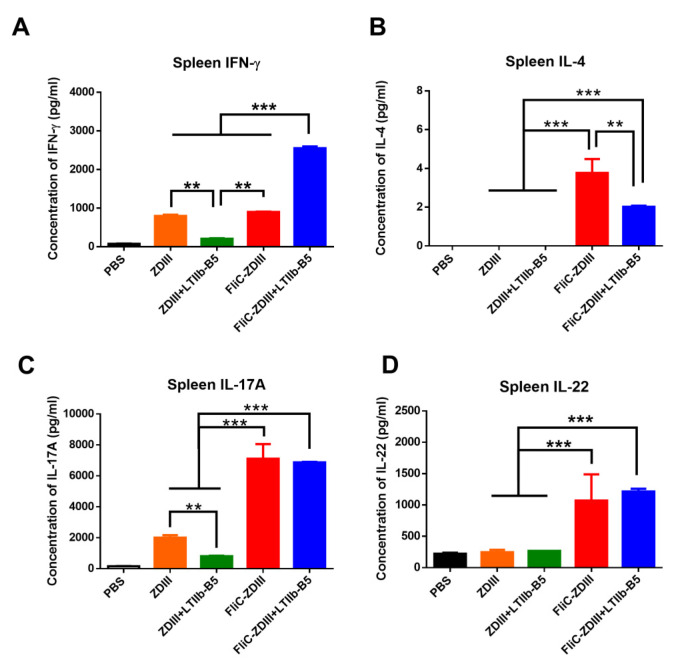
**ZDIII-specific T cell responses detected in spleen.** (**A**) IFN-γ production in stimulated cells from the spleen; (**B**) IL-4 production in stimulated cells from the spleen; (**C**) IL-17A production in stimulated cells from the spleen; (**D**) IL-22 production in stimulated cells from the spleen. Statistical test for multiple comparison for all groups except PBS were analyzed using one-way ANOVA with Tukey’s or Holm-Sidak’s multiple comparison tests. (**, *p* < 0.01 and ***, *p* < 0.001). Error bars are plotted as standard deviation from the mean value.

**Figure 4 pharmaceutics-14-01014-f004:**
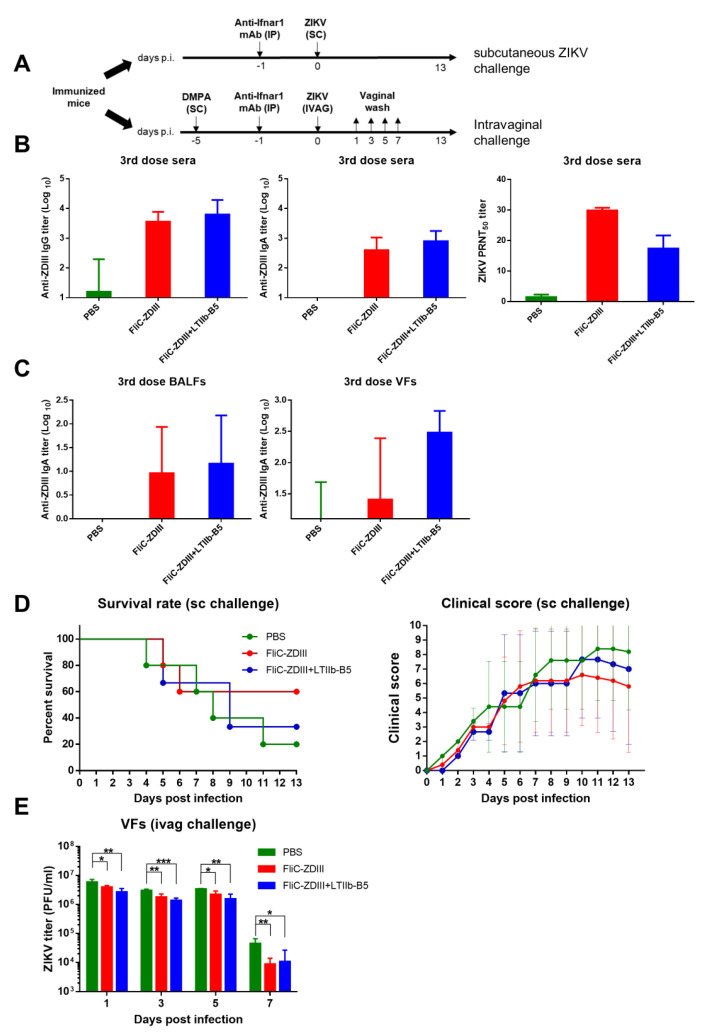
**Systemic and vaginal tract protection against ZIKV challenges in FliC-ZDIII-immunized mice.** (**A**) Groups of BALB/c mice (n = 5 per group) were intranasally immunized with PBS, 40 μg FliC-ZDIII, and 40 μg FliC-ZDIII + 3 μg LTIIb-B5 adjuvant for three doses in a 3-week interval. The sera, VFs, and spleens were collected at weeks 8 to determine anti-ZIKV antibody responses. The subcutaneous challenge groups received 2 mg of MAR1-5A3 mAb via the intraperitoneal route at 1 day before virus challenge with 10^8^ PFU of ZIKV. The intravaginal challenge groups were administrated with 2 mg DMPA at 5 days before virus challenge and 2 mg of MAR1-5A3 mAb via intraperitoneal route at 1 day before virus challenge with 10^8^ PFU of ZIKV; (**B**) Anti-ZDIII IgG, IgA and neutralizing antibody titers in sera; (**C**) Anti-ZDIII IgA titers in BALF and VFs; (**D**) the survival rate and clinical scores for the subcutaneous challenge were recorded for 14 days. Calculation of the clinical scores: normal = 0; ruffled fur = 2; lethargy, pinched, hunched, wasp-waisted = 3; labored breathing, rapid breathing, inactive, neurological = 5; and dead = 10; (**E**) the virus titers at days 1, 3, 5 and 7 post virus challenge were determined. Statistical test for multiple group comparison among the PBS, FliC-ZDIII, and FliC-ZDIII + LTIIb-B5 groups were analyzed using one-way ANOVA (and nonparametric) at days 1, 3, 5 and 7. (*, *p* < 0.05, **, *p* < 0.01 and ***, *p* < 0.001). Error bars are plotted as standard deviation from the mean value.

**Figure 5 pharmaceutics-14-01014-f005:**
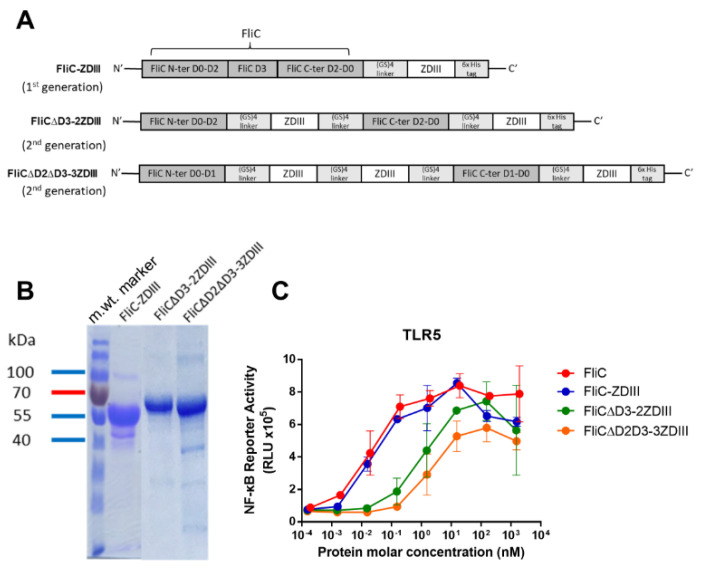
**Expression and characterization of second-generation FliCΔD3-2ZDIII and FliCΔD2ΔD3-3ZDIII antigens.** (**A**) FliCΔD3-2ZDIII and FliCΔD2ΔD3-3ZDIII were cloned into the pET-22b (+) vector for second-generation antigen expression; (**B**) FliCΔD3-2ZDIII and FliCΔD2ΔD3-3ZDIII were produced by *E. coli* and purified using nickel-chelated affinity chromatography. Purified proteins were verified by SDS-PAGE staining with Coomassie blue. The color ladder is a molecular weight marker; (**C**) TLR5 functionality in recombinant FliCΔD3-2ZDIII and FliCΔD2ΔD3-3ZDIII proteins were confirmed using the TLR5 assay. Error bars are plotted as standard deviation from the mean value.

**Figure 6 pharmaceutics-14-01014-f006:**
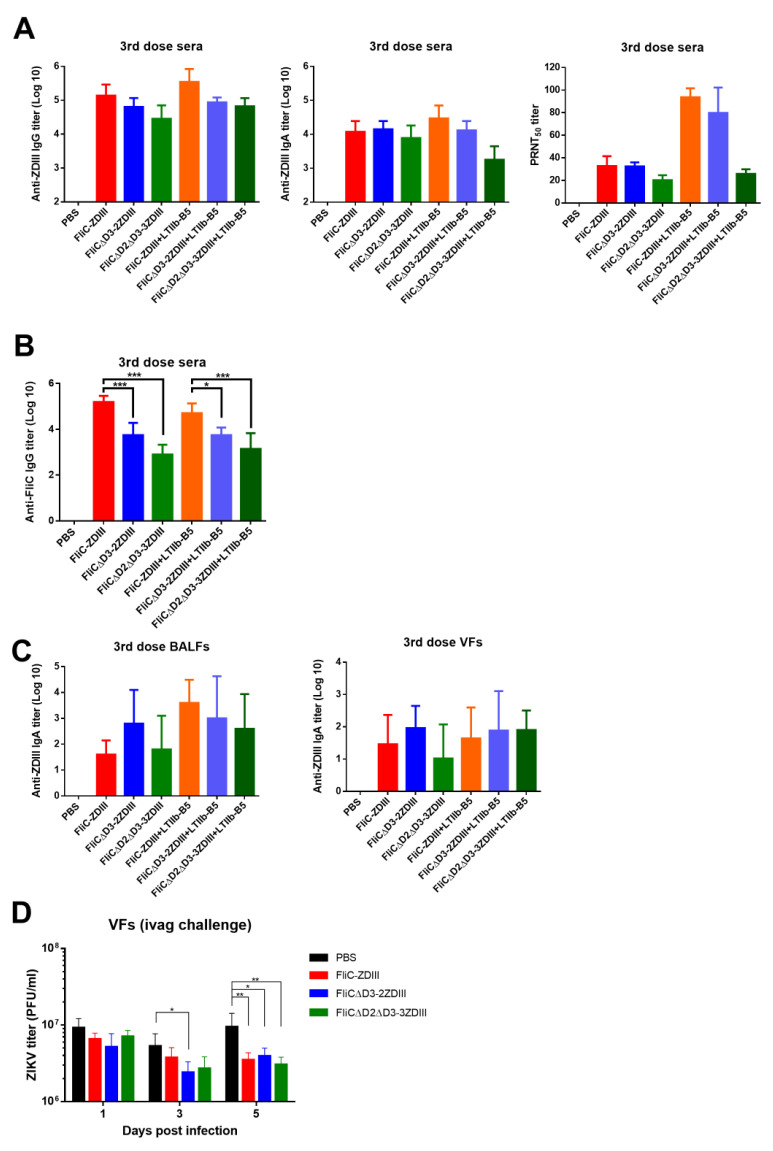
**Immune responses elicited by the second-generation FliCΔD3-2ZDIII and FliCΔD2ΔD3-3ZDIII antigens.** (**A**) Groups of BALB/c mice (n = 5 per group) were intranasally immunized with three doses of PBS, 40 μg FliC-ZDIII, 40 μg FliCΔD3-2ZDIII, 40 μg FliCΔD2ΔD3-3ZDIII, 40 μg FliC-ZDIII + 3 μg LTIIb-B5, 40 μg FliCΔD3-2ZDIII + 3 μg LTIIb-B5, and 40 μg FliCΔD2ΔD3-3ZDIII + 3 μg LTIIb-B5 in a three-week interval. Sera and VFs were collected at week 8; BALFs at week 9. The ZDIII-specific IgG, IgA and neutralizing antibody titers (PRNT-50) in sera, (**B**) Anti-FliC-specific IgG titers in sera, (**C**) ZDIII-specific IgA in BALFs and VFs. (**D**) Additional groups were conducted at week 9 for the intravaginal (IVAG) challenge of 10^8^ PFU of ZIKV. The virus titer in VFs was determined at day 1, 3 and 5 after the intravaginal (IVAG) challenge. Multiple group comparison for all groups not including the PBS-immunized group for (**A**–**C**). Multiple group comparison was conducted for all groups at the specific day (day 1, 3 and 5). Data were analyzed using one-way ANOVA with Tukey’s or Holm-Sidak’s multiple comparison tests. (*, *p* < 0.05, **, *p* < 0.01, and ***, *p* < 0.001). Error bars are plotted as standard deviation from the mean value.

**Figure 7 pharmaceutics-14-01014-f007:**
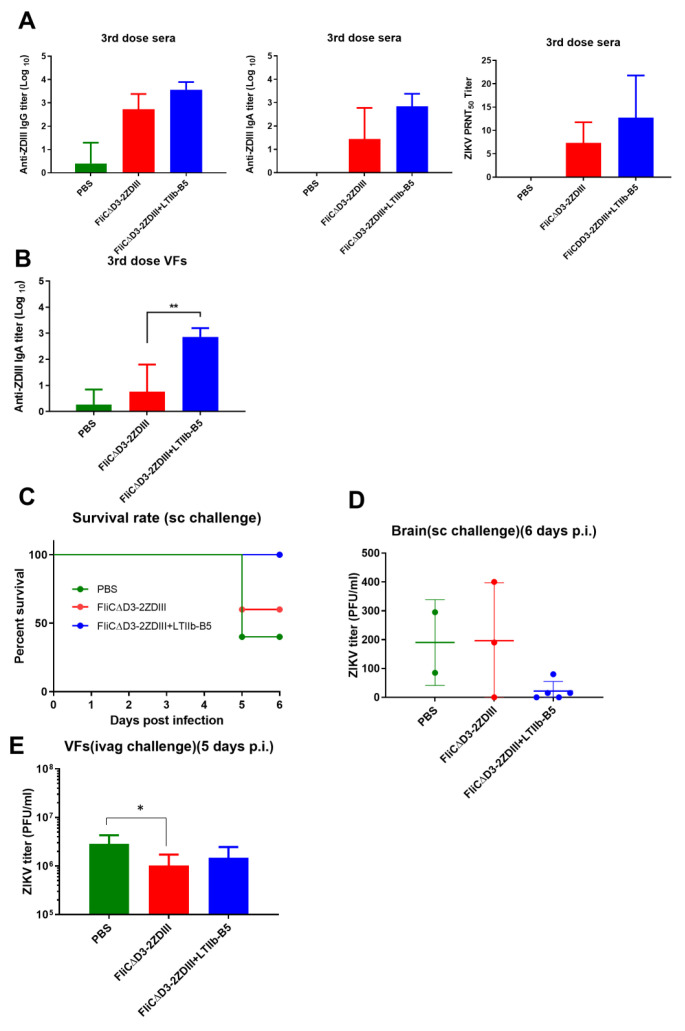
**Immune responses and protection of the second-generation FliCΔD3-2ZDIII antigen****plus LTIIb-B5 adjuvant.** (**A**) Groups of BALB/c mice (n = 5 per group) were intranasally immunized with (i) three doses of 40 μg FliCΔD3-2ZDIII, (ii) 40 μg FliCΔD3-2ZDIII plus LTIIb-B5 adjuvant, (iii) the first and second dose priming with 10^8^ pfu Ad-ZprME vector, followed by a third dose booster of 40 μg FliCΔD3-2ZDIII plus LTIIb-B5 adjuvant, alone with the PBS-immunized control group. Sera and VFs were collected at week 8. Subcutaneous (SC) or intravaginal (IVAG) challenges with 10^8^ PFU of ZIKV were conducted at week 9. The anti-ZDIII IgG, IgA, and neutralizing antibody titers (PRNT-50) in sera. (**B**) The anti-ZDIII IgA titer in VFs; (**C**) Survival rates recorded for 6 days; (**D**) The virus titer in brain at day 6 after SC challenge; (**E**) The virus titer in VFs determined at day 5 after IVAG challenge. Multiple group comparison was analyzed in all groups except the PBS group in (**A**,**B**). Multiple group comparison was analyzed in all groups. Data were analyzed using one-way ANOVA with Tukey’s or Holm-Sidak’s multiple comparison tests. (*, *p* < 0.05, **, *p* < 0.01). Error bars are plotted as standard deviation from the mean value.

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
