# Peer review of "Intranasal Immunization with Zika Virus Envelope Domain III-Flagellin Fusion Protein Elicits Systemic and Mucosal Immune Responses and Protection against Subcutaneous and Intravaginal Virus Challenges"

_pharmaceutics, 2022, doi:10.3390/pharmaceutics14051014_

Round 1

Reviewer 1 Report

This manuscript is a study to develop a mucosal subunit vaccine by conjugation with the well-known Flagellin adjuvant with Zika E protein DIII domain. It is well known that the mucosal vaccine using FliC induces an effective cellular immune response, this study did not show that such cell-mediated immunity also has an effective protective immune response.

Major comment

  1. For the vaginal colonization model, a mucosal immune response such as vaginal wash or saliva must be measured.
  2. It's not clear why the authors used ETEC toxin as an additional adjuvant. Although FliC induces an adjuvant effect through TLR5, the mechanism of the toxin is not clear, so adding it to this study may make the reader difficult to interpret the results. Could you delete Toxin part in the revised manuscript?
  3. Fig 1A needs to draw the protein in more detail and schematic. To show DIII, the amino acid number (such ad DIII102-203) must also be shown, and if a G54 linker is used, its sequence must also be shown. You also need to indicate where and how far FliC has been used.

Minor comment

  1. What is “The second-generation”? You should clarify First and Second generation ZDIII here. Meaning
  2. What is GS4 linker
  3. Fig 1C. Use molar concentration rather than ng/ml.
  4. It seems that the adverse effect of LTIIb is seen as the mice administered LTIIb died. It is necessary to explain this part or exclude the contents of LTIIb.
  5. I can't find the advantage of the 2nd generation vaccine. Clear immunological differences should be stated here. In addition, two statistic test should be performed “PBS vs. vaccine” and “1st generation vs. 2nd generation”.

Author Response

Reviewer#1

This manuscript is a study to develop a mucosal subunit vaccine by conjugation with the well-known Flagellin adjuvant with Zika E protein DIII domain. It is well known that the mucosal vaccine using FliC induces an effective cellular immune response, this study did not show that such cell-mediated immunity also has an effective protective immune response.

Response:

We have characterized the cellular immune response of ZDIII-specific CD4+ helper T cells of Th1, Th2, and Th17 response in the immunized mice (Fig. 3). The results indicated that intranasal immunization with FliC-ZDIII fusion protein induced potent Th1 and Th17 responses which may contribute to the production of mucosal IgA antibodies for the first line protection. We have not determined the CD8+ T cellular immunity which may also contribute to the protective immunity.

Major comment

  1. For the vaginal colonization model, a mucosal immune response such as vaginal wash or saliva must be measured.

Response:

We have measured the IgA antibodies in BALFs and VFs of the immunized mice in Fig. 2D, E; Fig. 4C; Fig. 6C; Fig. 7B and have characterize the elicited mucosal immune responses. However, due to the limited amount of VFs collected from the immunized mice after the vaginal virus challenges, we can only determine the virus titers in VFs at 1, 3, 5 days post infection (Figs. 4E, 6D, 7E) to assess the protection in the present work.

  1. It's not clear why the authors used ETEC toxin as an additional adjuvant. Although FliC induces an adjuvant effect through TLR5, the mechanism of the toxin is not clear, so adding it to this study may make the reader difficult to interpret the results. Could you delete Toxin part in the revised manuscript?

Response:

As we have previously shown the use of E. coli type IIb heat labile enterotoxin B subunit (LTIIb-B5) as a mucosal adjuvant for intranasal immunizations to enhance protective immunity against H5N1 avian influenza virus infection (Tang et al., 2020), we have included the use of LTIIb-B5 as an additional adjuvant in this work. Our findings have shown that the flagellin fusion antigen (FliC-ZDIII) alone did not elicit an effective mucosal IgA in VFs as well as the protection against intravaginal virus challenge (Fig. 7B, C, D). However, with the additional LTIIb-B5 adjuvant included in particularly for the second-generation FliCΔD3-2ZDIII fusion proteins, the combination formulation elicited the greatest protective immunity against subcutaneous and intravaginal ZIKV challenges (Fig. 7C, D, E).

  1. Fig 1A needs to draw the protein in more detail and schematic. To show DIII, the amino acid number (such ad DIII102-203) must also be shown, and if a G54 linker is used, its sequence must also be shown. You also need to indicate where and how far FliC has been used.

Response:

The cDNA from the ZDIII gene of the Brazil Paraiba_01/2015 strain (GI:1032590576, amino acid residues E589-697) and the fliC gene from Salmonella typhimurium (GI:217062, amino acid residues 1-495) were synthesized. The (GS)4 linker is the sequence of GSGSGSGS.

Minor comment

  1. What is “The second-generation”? You should clarify First and Second generation ZDIII here. Meaning

Response:

In the Result 3.4 section, “…….We further constructed the second-generation FliC-ZDIII fusion proteins by replacing FliC D3 with another ZDIII (FliCΔD3-2ZDIII), or replacing FliC D2 and D3 with two repeats (2x) of ZDIII (FliCΔD2ΔD3-3ZDIII) (Fig. 5A), as compared to FliC-ZDIII as the first generation antigen…..”

  1. What is GS4 linker

Response:

Corrected as a (GS)4 linker, the sequence of GSGSGSGS.

  1. Fig 1C. Use molar concentration rather than ng/ml.

Response:

Changed to molar concentration in Fig. 1C.

  1. It seems that the adverse effect of LTIIb is seen as the mice administered LTIIb died. It is necessary to explain this part or exclude the contents of LTIIb.

Response:

The adverse effect of LTIIb-B5 was shown in Fig. 4D for the survival rates and clinical scores of the FliC-ZDIII+LTIIb-B5 group compared to the FliC-ZDIII group following subcutaneous virus challenges. However, the adverse effect of LTIIb-B5 accounting for the differences are not statistically significant (Fig. 4D). The other experiments illustrated in Fig. 7C for intranasal immunization with the second-generation FliCΔD3-2ZDIII antigen without or with the additional use of LTIIb-B5 adjuvant, the survival rates for the FliCΔD3-2ZDIII + LTIIb-B5 group was 100% compared to 60% for the FliCΔD3-2ZDIII group and 40% for the PBS group (Fig. 7C). Notably, the numbers of immunized mice (n=5 per group) at 6 days after subcutaneous virus challenge dropped to n=2 for the PBS group and dropped to n=3 for the FliCΔD3-3ZDIII group. Therefore, it is unlikely that the adverse effect of LTIIb-B5 adjuvant is significant for this investigation in mice.

  1. I can't find the advantage of the 2nd generation vaccine. Clear immunological differences should be stated here. In addition, two statistic test should be performed “PBS vs. vaccine” and “1st generation vs. 2nd generation”.

Response:

The advantage of the 2nd generation vaccine FliCΔD3-2ZDIII is to reduce the elicitation of FliC-specific adaptive response without compromising the vaccine antigenicity. As the bacterial flagellin is an antigen commonly expressed by commensal and pathogenic bacteria in the guts, the flagellin-specific adaptive immunity may link to the development of chronic relapsing intestinal inflammation of the inflammatory bowel diseases such as Crohn's disease and ulcerative colitis [48-50]. Therefore, the 2nd generation vaccine has an inherent safety advantage with a reduced anti-flagellin antibody titer in sera (Fig. 6B) but still to elicit the same levels of anti-ZIKV IgG, IgA, and neutralizing antibodies in sera and anti-ZIKV IgA in BALFs and VFs similarly to the first generation FliC-ZDIII group (Fig. 6A, 6C). Therefore, the second-generation FliCΔD3-2ZDIII antigen can reduce the elicitation of FliC-specific adaptive response without compromising the vaccine antigenicity.

The statistic tests have been performed among the first generation FliC-ZDIII and two second generation FliCΔD3-2ZDIII FliCΔD2ΔD3-3ZDIII groups. Multiple group comparison for all groups (1st generation vs. 2nd generation) has not included the PBS-immunized group as stated in the legends of Fig. 6.

Reviewer 2 Report

The development of prophylactic vaccines against emerging infections is very important. We have faced this when COVID-19 pandemic was started. Humanity is not immune from the fact that once the Zika virus will mutated in such way that it acquires the potential of a pandemic virus. Therefore, this article is very important both for science and for practical health care. However, a minor revision of the manuscript is needed.

Point 1: Figure 2A, which illustrates methods of working with mice and is not related to specific results presented in Figure 2B-E, should be moved to Lines 149-152 as separate figure to better illustrate what was said in this paragraph. In addition, Figure 4A could be combined with Figure 2A under the title, for instance, “Test groups of mice…” or “Schedule of treatment of mice…,” etc.

Point 2: Materials and Methods. A description of mice supplier, their housing, husbandry and pain management is missing in the manuscript. How the mice were sacrificed at the end of the experiment?

Point 3: Did the authors pretreat mice sera somehow to remove non-specific inhibitors?

Point 4: Figure 1B. Please indicate that the color “ladder” is a molecular weight marker.

Point 5: Figure 1C, axis X. Please, specify what concentration are you referring to.

Point 6: Figure 5B. Please indicate that the color “ladder” is a molecular weight marker.

Point 7: Figure 5C, axis X. Please, specify what concentration are you referring to.

Point 8: Discussion. As of to date, no genetically engineered vaccine has proven to be sufficiently immunogenic. According to the presented results, the Zika virus intranasal vaccine developed by the authors stimulates a pronounced immune response. If the authors, even in one sentence/paragraph, compared the intensity of the immune response to any modern synthetic vaccines and to their own vaccine, this would greatly decorate the article.

Point 9: According to the instructions for authors of Pharmaceutics, the Conclusions section is not mandatory but if it will be added to the manuscript then it will look more finished.

Author Response

Reviewer#2

The development of prophylactic vaccines against emerging infections is very important. We have faced this when COVID-19 pandemic was started. Humanity is not immune from the fact that once the Zika virus will mutated in such way that it acquires the potential of a pandemic virus. Therefore, this article is very important both for science and for practical health care. However, a minor revision of the manuscript is needed.

Response:

Thanks for the reviewer’s endorsement and we have made the revision according to the comments.

Point 1: Figure 2A, which illustrates methods of working with mice and is not related to specific results presented in Figure 2B-E, should be moved to Lines 149-152 as separate figure to better illustrate what was said in this paragraph. In addition, Figure 4A could be combined with Figure 2A under the title, for instance, “Test groups of mice…” or “Schedule of treatment of mice…,” etc.

Response:

We would prefer not separating Fig. 2A and Fig. 4A into another figure(s) as the reviewer suggested in the Materials and Methods section. We think the illustration of the methods of working with mice for immunization and challenge can give a more coherent presentation with the combination with the experimental results in each of the specified experiments.

Point 2: Materials and Methods. A description of mice supplier, their housing, husbandry and pain management is missing in the manuscript. How the mice were sacrificed at the end of the experiment?

Response:

BALB/C mice were provided by National Laboratory Animal Center and kept in plastic see-through cages. Each cage contains a water bottle, feeder and corn cob bedding. Environment condition was based on IACUC policy (Condition: 07~19 light & 19~07 dark; Temp 24 oC). Mice were euthanized after the completion of experiments. Pain of animals were minimized by agents (Ketoprofen). Sacrifice of mice was carried out by CO2 inhalation.

Point 3: Did the authors pretreat mice sera somehow to remove non-specific inhibitors?

Response:

Antisera were pretreated with heat inactivation at 56°C for 30 min for complement inactivation.

Point 4: Figure 1B. Please indicate that the color “ladder” is a molecular weight marker.

Response:

The molecular weight marker is indicated as the color ladder in Figure 1B.

Point 5: Figure 1C, axis X. Please, specify what concentration are you referring to.

Response:

Protein concentrations for FliC and FliC-ZDIII. Deleted the PBS.

Point 6: Figure 5B. Please indicate that the color “ladder” is a molecular weight marker.

Response:

The molecular weight (m.wt.) markers indicated as the color ladder in Figure 1B.

Point 7: Figure 5C, axis X. Please, specify what concentration are you referring to.

Response:

Protein concentration and deleted the PBS.

Point 8: Discussion. As of to date, no genetically engineered vaccine has proven to be sufficiently immunogenic. According to the presented results, the Zika virus intranasal vaccine developed by the authors stimulates a pronounced immune response. If the authors, even in one sentence/paragraph, compared the intensity of the immune response to any modern synthetic vaccines and to their own vaccine, this would greatly decorate the article.

Response:

We have added the following sentences in the third paragraph in the Discussion section.

“….Compared to the results that we previously reported using FliC-ZDIII for two-dose intramuscular immuniztion or the first dose adenovirus vector-priming and the second dose FliC-ZDIII booster in BALB/c mice (33), the three-dose intranasal immunization with FliC-ZDIII and LTIIb-B5 adjuvant elicited higher titers of neutralizing antibodies in sera (Figs. 2B, 4B, 6A, 7A). Therefore, intranasal immunization with FliC-ZDIII fusion protein for a three-dose regimen can elicit a comparable systematic immune response similarly to intramuscular immunization. To our knowledge, this is the first report using mucosal delivery of ZIKV recombinant vaccines…”

Point 9: According to the instructions for authors of Pharmaceutics, the Conclusions section is not mandatory but ifit will be added to the manuscript then it will look more finished.

Response:

The conclusion section has been added to meet the journal’s style for publication.

Round 2

Reviewer 1 Report

Some questions have not been answered or corrected in the revised manuscript, but important parts were sufficiently reflected.

Author Response

Response:

We have added the comments in the end of the 4th paragraph in the Discussion section as following: “….Moreover, we did not measure the CTL response elicited by intranasal immunizations with FliC-ZDIII plus LTIIb-B5 adjuvant in the present study. Further examination is needed to determine the cell-mediated immunity may also contribute to an effective protective immune response.”